# Adversarially Robust Learning with Tolerance

## Abstract

We initiate the study of tolerant adversarial PAC learning with respect to metric perturbation sets. In adversarial PAC learning, an adversary is allowed to replace a test point $x$ with an arbitrary point in a closed ball of radius $r$ centered at $x$. In the tolerant version, the error of the learner is compared with the best achievable error with respect to a slightly larger perturbation radius $(1 + \gamma)r$. This simple tweak helps us bridge the gap between theory and practice and obtain the first PAC-type guarantees for algorithmic techniques that are popular in practice. Furthermore, our sample complexity bounds improve exponentially over best known (non-tolerant) bounds in terms of the VC dimension of the hypothesis class. In particular, for perturbation sets with doubling dimension $d$, we show that a variant of the "perturb-and-smooth" algorithm PAC learns any hypothesis class $\mathcal{H}$ with VC dimension $v$ in the $\gamma$-tolerant adversarial setting with $O\left(\frac{v(1+1/\gamma)^{O(d)}}{\varepsilon}\right)$ samples. This guarantee holds in the tolerant robust realizable setting. We extend this to the agnostic case by designing a novel sample compression scheme based on the perturb-and-smooth approach. This compression-based algorithm has a linear dependence on the doubling dimension as well as the VC-dimension.

## 1 Introduction

Several empirical studies (Szegedy et al., 2014; Goodfellow et al., 2018) have demonstrated that models trained to have a low accuracy on a data set often have the undesirable property that a small perturbation to an input instance can change the label outputted by the model. For most domains this does not align with human intuition and thus indicates that the learned models are not representing the ground truth despite obtaining good accuracy on test sets.

The theory of PAC-learning characterizes the conditions under which learning is possible. For binary classification, the following conditions are sufficient: a) unseen data should arrive from the same distribution as training data, and b) the class of models should have a low capacity (as measured, for example, by its VC dimension). If these conditions are met, an *Empirical Risk Minimizer* (ERM) that simply optimizes model parameters to maximize accuracy on the training set learns successfully.

Recent work has studied test-time adversarial perturbations under the PAC-learning framework. If an adversary is allowed to perturb data during test time then the conditions above do not hold, and we cannot hope for the model to learn to be robust just by running ERM. Thus, the goal here is to bias the learning process towards finding models where label-changing perturbations are rare. This is achieved by defining a loss function that combines both classification error and the probability of seeing label-changing perturbations, and learning models that minimize this loss on unseen data. It has been shown that even though (robust) ERM can fail in this setting, PAC learning is still possible as long as we know during training the kinds of perturbations we want to guard against at test time (Montasser et al., 2019). This result holds for all perturbation sets. However, the learning

algorithm is significantly more complex than robust ERM and requires a large number of samples (with the best known sample complexity bounds potentially being exponential in the VC-dimension).

We study a *tolerant* version of the adversarially robust learning framework and restrict the perturbations to balls in a general metric space with finite doubling dimension. We show this slight shift in the learning objective yields significantly improved sample complexity bounds through a simpler learning paradigm than what was previously known. In fact, we show that a version of the common "perturb-and-smooth" paradigm successfully PAC-learns any class of bounded VC dimension in this setting.

**Learning in general metric spaces.** What kinds of perturbations should a learning algorithm guard against? Any transformation of the input that we believe should not change its label could be a viable perturbation for the adversary to use. The early works in this area considered perturbations contained within a small $\ell_p$-ball of the input. More recent work has considered other transformations such as a small rotation, or translation of an input image (Engstrom et al., 2019; Fawzi & Frossard, 2015; Kanbak et al., 2018; Xiao et al., 2018), or even adding small amounts of fog or snow (Kang et al., 2019). It has also been argued that small perturbations in some *feature space* should be allowed as opposed to the input space (Inkawhich et al., 2019; Sabour et al., 2016; Xu et al., 2020; Song et al., 2018; Hosseini & Poovendran, 2018). This motivates the study of more general perturbations.

We consider a setting where the input comes from a domain that is equipped with a distance metric and allows perturbations to be within a small metric ball around the input. Earlier work on general perturbation sets (for example, (Montasser et al., 2019)) considered arbitrary perturbations. In this setting one does not quantify the magnitude of a perturbation and thus cannot talk about small versus large perturbations. Modeling perturbations using a metric space enables us to do that while also keeping the setup general enough to be able to encode a large variety of perturbation sets by choosing appropriate distance functions.

**Learning with tolerance.** In practice, we often believe that small perturbations of the input should not change its label but we do not know *precisely* what small means. However, in the PAC-learning framework for adversarially robust classification, we are required to define a precise perturbation set and learn a model that has error arbitrarily close to the smallest error that can be achieved with respect to that perturbation set. In other words, we aim to be arbitrarily close to a target that was picked somewhat arbitrarily to begin with. Due to the uncertainty about the correct perturbation size, it is more meaningful to allow for a wider range of error values. To achieve this, we introduce the concept of tolerance. In the tolerant setting, in addition to specifying a perturbation size $r$, we introduce a tolerance parameter $\gamma$ that encodes our uncertainty about the size of allowed perturbations. Then, for any given $\epsilon > 0$, we aim to learn a model whose error with respect to perturbations of size $r$ is at most $\epsilon$ more than the smallest error achievable with respect to perturbations of size $r(1 + \gamma)$.

## 2 Our results

In this paper we formalize and initiate the study of the problem of adversarially robust learning in the tolerant setting for general metric spaces and provide two algorithms for the task. Both of our algorithms rely on: 1) modifying the training data by randomly sampling points from the perturbation sets around each data point, and 2) smoothing the output of the model by taking a majority over the labels returned by the model for nearby points.

Our first algorithm starts by modifying the training set by randomly perturbing each training point using a certain distribution (see Section 5 for details). It then trains a (non-robust) PAC learner (such as ERM) on the perturbed training set to find a hypothesis $h$. Finally, it outputs a smooth version of $h$. The smoothing step replaces $h(x)$ at each point $x$ with the a majority label outputted by $h$ on the points around $x$. We show that for metric spaces of a fixed doubling dimension, this algorithm successfully learns in the (robustly realizable) tolerant setting.

**Theorem 1** (Informal version of Theorem 10). *Let $(X, \mathrm{dist})$ be a metric space with doubling dimension $d$ and $\mathcal{H}$ a hypothesis class. Assuming robust realizability, $\mathcal{H}$ can be learned tolerantly in the adversarially robust setting using $O\left(\frac{(1+1/\gamma)^{O(d)}\mathrm{VC}(\mathcal{H})}{\epsilon}\right)$ samples, where $\gamma$ encodes the amount of allowed tolerance, and $\epsilon$ is the desired accuracy.*

An interesting feature of the above result is the linear dependence of the sample complexity with respect to $\mathrm{VC}(\mathcal{H})$. This is in contrast to the best known upper bound for non-tolerant adversarial setting (Montasser et al., 2019) which depends on the *dual VC dimension* of the hypothesis class and in general is exponential in $\mathrm{VC}(\mathcal{H})$. Moreover, this is the first PAC type guarantee for the general perturb-and-smooth paradigm, indicating that the tolerant adversarial learning is the "right" learning model for studying these approaches. While the above method enjoys simplicity and can be computationally efficient, one downside is that its sample complexity grows exponentially with the doubling dimension. For instance, such algorithm cannot be used on high-dimensional data in the Euclidean space. Another limitation is that the guarantee holds only in the (robustly) realizable setting. We propose another algorithm that improves the dependence on doubling dimension, and works in the general agnostic setting.

**Theorem 2** (Informal version of Corollary 16). *Let $(X, \mathrm{dist})$ be a metric space with doubling dimension $d$ and $\mathcal{H}$ a hypothesis class. Then $\mathcal{H}$ can be learned tolerantly in the adversarially robust setting using $\widetilde{O}\left(\frac{O(d)\mathrm{VC}(\mathcal{H})\log(1+1/\gamma)}{\epsilon^2}\right)$ samples, where $\widetilde{O}$ hides logarithmic factors, $\gamma$ encodes the amount of allowed tolerance, and $\epsilon$ is the desired accuracy.*

This algorithm exploits the connection between sample compression and adversarially robust learning Montasser et al. (2019). However, unlike Montasser et al. (2019), our new compression scheme sidesteps the dependence on the dual VC dimension. As a result, we get an exponential improvement over the best known (nontolerant) sample complexity in terms of dependence on VC dimension.

# 3 Related work

PAC-learning for adversarially robust classification has been studied extensively in recent years (Cullina et al., 2018; Awasthi et al., 2019; Montasser et al., 2019; Feige et al., 2015; Attias et al., 2019; Montasser et al., 2020a; Ashtiani et al., 2020). These works provide learning algorithms that guarantee low generalization error in the presence of adversarial perturbations in various settings. The most general result is due to (Montasser et al., 2019) which is proved for general hypothesis classes and perturbation sets. All of the above results assume that the learner knows the kinds of perturbations allowed for the adversary. Some more recent papers have considered scenarios where the learner does not even need to know that. Goldwasser et al. (2020) allow the adversary to perturb test data in unrestricted ways and are still able to provide learning guarantees. The catch is that it only works in the transductive setting and only if the learner is allowed to abstain from making a prediction on some test points. Montasser et al. (2021a) consider the case where the learner needs to infer the set of allowed perturbations by observing the actions of the adversary.

Tolerance was introduced by Ashtiani et al. (2020) but in the context of certification. They provide examples where certification is not possible unless we allow some tolerance. Montasser et al. (2021b) study transductive adversarial learning and provide a "tolerant" guarantee. Note that unlike our work, the main focus of this paper is on the transductive setting. Moreover, they do not specifically study tolerance with respect to metric perturbation sets. Without a metric, it is not meaningful to expand perturbation sets by a factor $(1 + \gamma)$ (as we do in the our definition of tolerance). Instead, they expand their perturbation sets by applying two perturbations in succession, which is akin to setting $\gamma = 1$. In contrast, our results hold in the more common inductive setting, and capture a more realistic setting where $\gamma$ is any (small) real number larger than zero.

Like many recent adversarially robust learning algorithms (Feige et al., 2015; Attias et al., 2019), our first algorithm relies on calls to a non-robust PAC-learner. Montasser et al. (2020b) formalize the question of reducing adversarially robust learning to non-robust learning and study finite perturbation sets of size $k$. They show a reduction that makes $O(\log^2 k)$ calls to the non-robust learner and also prove a lower bound of $\Omega(\log k)$. It will be interesting to see if our algorithms can be used to obtain better bounds for the tolerant setting. Our first algorithm makes one call to the non-robust PAC-learner at training time, but needs to perform potentially expensive smoothing for making actual predictions (see Theorem 10).

The techniques of randomly perturbing the training data and smoothing the output classifier has been extensively used in practice and has shown good empirical success. Augmenting the training data with some randomly perturbed samples was used for handwriting recognition as early as in (Yaeger et al., 1996). More recently, "stability training" was introduced in (Zheng et al., 2016) for state of the art

image classifiers where training data is augmented with Gaussian perturbations. Empirical evidence was provided that the technique improved the accuracy against naturally occurring perturbations. Augmentations with non-Gaussian perturbations of a large variety were considered in (Hendrycks et al., 2019).

Smoothing the output classifier using random samples around the test point is a popular technique for producing *certifiably* robust classifiers. A certification, in this context, is a guarantee that given a test point $x$, all points within a certain radius of $x$ receive the same label as $x$. Several papers have provided theoretical analyses to show that smoothing produces certifiably robust classifiers (Cao & Gong, 2017; Cohen et al., 2019; Lecuyer et al., 2019; Li et al., 2019; Liu et al., 2018; Salman et al., 2019; Levine & Feizi, 2020).

However, to the best of our knowledge, a PAC-like guarantee has not been shown for any algorithm that employs training data perturbations or output classifier smoothing, and our paper provides the first such analysis.

# 4 Notations and setup

We denote by $X$ the input domain and by $Y = \{0, 1\}$ the binary label space. We assume that $X$ is equipped with a metric dist. A hypothesis $h : X \to Y$ is a function that assigns a label to each point in the domain. A hypothesis class $\mathcal{H}$ is a set of such hypotheses. For a sample $S = ((x_1, y_1), \ldots, (x_n, y_n)) \in (X \times Y)^n$, we use the notation $S^X = \{x_1, x_2, \ldots, x_n\}$ to denote the collection of domain points $x_i$ occurring in $S$. The binary (also called 0-1) loss of $h$ on data point $(x, y) \in X \times Y$ is defined by

$$\ell^{0/1}(h, x, y) = \mathbb{1}\left[h(x) \neq y\right],$$

where $\mathbb{1}[.]$ is the indicator function. Let $P$ by a probability distribution over $X \times Y$. Then the *expected binary loss* of $h$ with respect to $P$ is defined by

$$\mathcal{L}_P^{0/1}(h) = \mathbb{E}_{(x,y)\sim P}[\ell^{0/1}(h, x, y)]$$

Similarly, the *empirical binary loss* of $h$ on sample $S = ((x_1, y_1), \ldots, (x_n, y_n)) \in (X \times Y)^n$ is defined as $\mathcal{L}_S^{0/1}(h) = \frac{1}{n}\sum_{i=1}^{n} \ell^{0/1}(h, x_i, y_i)$. We also define the *approximation error* of $\mathcal{H}$ with respect to $P$ as $\mathcal{L}_P^{0/1}(\mathcal{H}) = \inf_{h \in \mathcal{H}} \mathcal{L}_P^{0/1}(h)$.

A *learner* $\mathcal{A}$ is a function that takes in a finite sequence of labeled instances $S = ((x_1, y_1), \ldots, (x_n, y_n))$ and outputs a hypothesis $h = \mathcal{A}(S)$. The following definition abstracts the notion of PAC learning Vapnik & Chervonenkis (1971); Valiant (1984).

**Definition 3** (PAC Learner). *Let $\mathcal{P}$ be a set of distributions over $X \times Y$ and $\mathcal{H}$ a hypothesis class. We say $\mathcal{A}$ PAC learns $(\mathcal{H}, \mathcal{P})$ with $m_{\mathcal{A}} : (0, 1)^2 \to \mathbb{N}$ samples if the following holds: for every distribution $P \in \mathcal{P}$ over $X \times Y$, and every $\epsilon, \delta \in (0, 1)$, if $S$ is an i.i.d. sample of size at least $m_{\mathcal{A}}(\epsilon, \delta)$ from $P$, then with probability at least $1 - \delta$ (over the randomness of $S$) we have*

$$\mathcal{L}_P(\mathcal{A}(S)) \leq \mathcal{L}_P(\mathcal{H}) + \epsilon.$$

*$\mathcal{A}$ is called an* agnostic learner *if $\mathcal{P}$ is the set of all distributions on $X \times Y$, and a* realizable learner *if $\mathcal{P} = \{P : \mathcal{L}_P(\mathcal{H}) = 0\}$.*

The smallest function $m : (0, 1)^2 \to \mathbb{N}$ for which there exists a learner $\mathcal{A}$ that satisfies the above definition with $m_{\mathcal{A}} = m$ is referred to as the (realizable or agnostic) *sample complexity* of learning $\mathcal{H}$.

The existence of sample-efficient PAC learners for VC classes is a standard result Vapnik & Chervonenkis (1971). We state the results formally in Appendix A.

## 4.1 Tolerant adversarial PAC learning

Let $\mathcal{U} : X \to 2^X$ be a function that maps each point in the domain to the set of its "admissible" perturbations. We call this function the *perturbation type*. The adversarial loss of $h$ with respect to $\mathcal{U}$ on $(x, y) \in X \times Y$ is defined by

$$\ell^{\mathcal{U}}(h, x, y) = \max_{z \in \mathcal{U}(x)} \{\ell^{0/1}(h, z, y)\}$$

The *expected adversarial loss* with respect to $P$ is defined by $\mathcal{L}_P^{\mathcal{U}}(h) = \mathbb{E}_{(x,y)\sim P}\ell^{\mathcal{U}}(h,x,y)$. The *empirical adversarial loss* of $h$ on sample $S = ((x_1, y_1), \ldots, (x_n, y_n)) \in (X \times Y)^n$ is defined by $\mathcal{L}_S^{\mathcal{U}}(h) = \frac{1}{n}\sum_{i=1}^n \ell^{\mathcal{U}}(h, x_i, y_i)$. Finally, the *adversarial approximation error* of $\mathcal{H}$ with respect to $\mathcal{U}$ and $P$ is defined by $\mathcal{L}_P^{\mathcal{U}}(\mathcal{H}) = \inf_{h \in \mathcal{H}} \mathcal{L}_P^{\mathcal{U}}(h)$.

The following definition generalizes the setting of PAC adversarial learning to what we call the *tolerant* setting, where we consider two perturbation types $\mathcal{U}$ and $\mathcal{V}$. We say $\mathcal{U}$ is *contained in* $\mathcal{V}$ and and write it as $\mathcal{U} \prec \mathcal{V}$ if $\mathcal{U}(x) \subset \mathcal{V}(x)$ for all $x \in X$.

**Definition 4** (Tolerant Adversarial PAC Learner). *Let $\mathcal{P}$ be a set of distributions over $X \times Y$, $\mathcal{H}$ a hypothesis class, and $\mathcal{U} \prec \mathcal{V}$ two perturbation types. We say $\mathcal{A}$ tolerantly PAC learns $(\mathcal{H}, \mathcal{P}, \mathcal{U}, \mathcal{V})$ with $m_{\mathcal{A}} : (0,1)^2 \to \mathbb{N}$ samples if the following holds: for every distribution $P \in \mathcal{P}$ and every $\epsilon, \delta \in (0,1)$, if $S$ is an i.i.d. sample of size at least $m_{\mathcal{A}}(\epsilon, \delta)$ from $P$, then with probability at least $1 - \delta$ (over the randomness of $S$) we have*

$$\mathcal{L}_P^{\mathcal{U}}(\mathcal{A}(S)) \leq \mathcal{L}_P^{\mathcal{V}}(\mathcal{H}) + \epsilon.$$

*We say $\mathcal{A}$ is a tolerant PAC learner in the* agnostic *setting if $\mathcal{P}$ is the set of all distributions over $X \times Y$, and in the* tolerantly realizable *setting if $\mathcal{P} = \{P : \mathcal{L}_P^{\mathcal{V}}(\mathcal{H}) = 0\}$.*

In the above context, we refer to $\mathcal{U}$ as the *actual perturbation type* and to $\mathcal{V}$ as the *reference perturbation type*. The case where $\mathcal{U}(x) = \mathcal{V}(x)$ for all $x \in X$ corresponds to the usual adversarial learning scenario (with no tolerance).

## 4.2 Tolerant adversarial PAC learning in metric spaces

If $X$ is equipped with a metric $\mathrm{dist}(.,.)$, then $\mathcal{U}(x)$ can be naturally defined by a ball of radius $r$ around $x$, i.e., $\mathcal{U}(x) = \mathcal{B}_r(x) = \{z \in X \mid \mathrm{dist}(x,z) \leq r\}$. To simplify the notation, we sometimes use $\ell^r(h,x,y)$ instead of $\ell^{\mathcal{B}_r}(h,x,y)$ to denote the adversarial loss with respect to $\mathcal{B}_r$.

In the tolerant setting, we consider the perturbation sets $\mathcal{U}(x) = \mathcal{B}_r(x)$ and $\mathcal{V}(x) = \mathcal{B}_{(1+\gamma)r}(x)$, where $\gamma > 0$ is called the *tolerance parameter*. Note that $\mathcal{U} \prec \mathcal{V}$. We now define PAC learning with respect to the metric space.

**Definition 5** (Tolerant Adversarial Learning in metric spaces). *Let $(X, \mathrm{dist})$ be a metric space, $\mathcal{H}$ a hypothesis class, and $\mathcal{P}$ a set of distributions of $X \times Y$. We say $(\mathcal{H}, \mathcal{P}, \mathrm{dist})$ is tolerantly PAC learnable with $m : (0,1)^3 \to \mathbb{N}$ samples when for every $r, \gamma > 0$ there exist a PAC learner $\mathcal{A}_{r,\gamma}$ for $(\mathcal{H}, \mathcal{P}, B_r, B_{r(1+\gamma)})$ that uses $m(\epsilon, \delta, \gamma)$ samples.*

**Remark 6.** *In this definition the learner receives $\gamma$ and $r$ as input but its sample complexity does not depend on $r$ (but can depend on $\gamma$). Also, as in Definition 4, the tolerantly realizable setting corresponds to $\mathcal{P} = \{P : \mathcal{L}_P^{r(1+\gamma)}(\mathcal{H}) = 0\}$ while in the agnostic setting $\mathcal{P}$ is the set of all distributions over $X \times Y$.*

The doubling dimension and the doubling measure of the metric space will play important roles in our analysis. We refer the reader to Appendix B for their definitions.

We will use the following lemma in our analysis, whose proof can be found in Appendix B:

**Lemma 7.** *For any family $\mathcal{M}$ of complete, doubling metric spaces, there exist constants $c_1, c_2 > 0$ such that for any metric space $(X, \mathrm{dist}) \in \mathcal{M}$ with doubling dimension $d$, there exists a measure $\mu$ such that if a ball $\mathcal{B}_r$ of radius $r > 0$ is completely contained inside a ball $\mathcal{B}_{\alpha r}$ of radius $\alpha r$ (with potentially a different center) for any $\alpha > 1$, then $0 < \mu(\mathcal{B}_{\alpha r}) \leq (c_1\alpha)^{c_2 d}\mu(\mathcal{B}_r)$. Furthermore, if we have a constant $\alpha_0 > 1$ such that we know that $\alpha \geq \alpha_0$ then the bound can be simplified to $0 < \mu(\mathcal{B}_{\alpha r}) \leq \alpha^{\zeta d}\mu(\mathcal{B}_r)$, where $\zeta$ depends on $\mathcal{M}$ and $\alpha_0$.*

Later, we will set $\alpha = 1 + 1/\gamma$ where $\gamma$ is the tolerance parameter. Since we are mostly interested in small values of $\gamma$, suppose we decide on some loose upper bound $\Gamma \gg \gamma$. This corresponds to saying that there exists some $\alpha_0 > 1$ such that $\alpha \geq \alpha_0$.

It is worth noting that in the special case of Euclidean metric spaces, we can set both $c_1$ and $c_2$ to be 1. In the rest of the paper, we will assume we have a loose upper bound $\Gamma \gg \gamma$ and use the simpler bound from Lemma 24 extensively.

Given a metric space $(X, d)$ and a measure $\mu$ defined over it, for any subset $Z \subseteq X$ for which $\mu(Z)$ is non-zero and finite, $\mu$ induces a *probability* measure $P_Z^{\mu}$ over $Z$ as follows. For any set $Z' \subseteq Z$ in

the $\sigma$-algebra over $Z$, we define $P_Z^\mu(Z') = \mu(Z')/\mu(Z)$. With a slight abuse of notation, we write $z \sim Z$ to mean $z \sim P_Z^\mu$ whenever we know $\mu$ from the context.

Our learners rely on being able to sample from $P_Z^\mu$. Thus we define the following oracle, which can be implemented efficiently for $\ell_p$ spaces.

**Definition 8** (Sampling Oracle). *Given a metric space $(X, \mathrm{dist})$ equipped with a doubling measure $\mu$, a sampling oracle is an algorithm that when queried with a $Z \subseteq X$ such that $\mu(Z)$ is finite, returns a sample drawn from $P_Z^\mu$. We will use the notation $z \sim Z$ for queries to this oracle.*

## 5 The perturb-and-smooth approach for tolerant adversarial learning

In this section we focus on tolerant adversarial PAC learning in metric spaces (Definition 5), and show that VC classes are tolerantly PAC learnable in the tolerantly realizable setting. Interestingly, we prove this result using an approach that resembles the "perturb-and-smooth" paradigm which is used in practice (for example, (Cohen et al., 2019). The overall idea is to "perturb" each training point $x$, train a classifier on the "perturbed" points, and "smooth out" the final hypothesis using a certain majority rule.

For this, we employ three perturbation types: $\mathcal{U}$ and $\mathcal{V}$ play the role of the *actual* and the *reference* perturbation type respectively. Additionally, we consider a perturbation type $\mathcal{W} : X \to 2^X$, which is used for smoothing. We assume $\mathcal{U} \prec \mathcal{V}$ and $\mathcal{W} \prec \mathcal{V}$. For this section, we will use metric balls for the three types. Specifically, if $\mathcal{U}$ consists of balls of radius $r$ for some $r > 0$, then $\mathcal{W}$ will consists of balls of radius $\gamma r$ and $\mathcal{V}$ will consist of balls of radius $(1 + \gamma)r$.

**Definition 9** (Smoothed classifier). *For a hypothesis $h : X \to \{0, 1\}$, we let $\bar{h}_{\mathcal{W}}$ denote the classifier resulting from replacing the label $h(x)$ with the average label over $\mathcal{W}(x)$, that is*

$$\bar{h}_{\mathcal{W}}(x) = \mathbb{1}\left[\mathbb{E}_{x' \sim \mathcal{W}(x)} h(x') \geq 1/2\right]$$

*For metric perturbation types, where $\mathcal{W}$ is a ball of some radius $r$, we also use the notation $\bar{h}_r$ and when the type $\mathcal{W}$ is clear from context, we may omit the subscript altogether and simply write $\bar{h}$ for the smoothed classifier.*

**The tolerant perturb-and-smooth algorithm** We propose the following learning algorithm, TPaS, for tolerant learning in metric spaces. Let the perturbation radius be $r > 0$ for the actual type $\mathcal{U} = \mathcal{B}_r$, and let $S = ((x_1, y_1), \ldots, (x_m, y_m))$ be the training sample. For each $x_i \in S^X$, the learner samples a point $x_i' \sim \mathcal{B}_{r \cdot (1+\gamma)}(x_i)$ (using the sampling oracle) from the expanded reference perturbation set $\mathcal{V}(x_i) = \mathcal{B}_{(1+\gamma)r}(x_i)$. Let $S' = ((x_1', y_1), \ldots, (x_m', y_m))$. TPaS then invokes a (standard, non-robust) PAC learner $\mathcal{A}_{\mathcal{H}}$ for the hypothesis class $\mathcal{H}$ on the perturbed data $S'$. We let $\hat{h} = \mathcal{A}_{\mathcal{H}}(S')$ denote the output of this PAC learner. Finally, TPaS outputs the $\mathcal{W}$-smoothed version of $\bar{h}_{\gamma r}$ for $\mathcal{W} = \mathcal{B}_{\gamma r}$. That is, $\bar{h}_{\gamma r}(x)$ is simply the majority label in a ball of radius $\gamma r$ around $x$ with respect to the distribution defined by $\mu$, see also Definition 9. We will prove below that this $\bar{h}_{\gamma r}$ has a small $\mathcal{U}$-adversarial loss. Algorithm 1 below summarizes our learning procedure.

---

**Algorithm 1** Tolerant Perturb and Smooth (TPaS)

---

**Input:** Radius $r$, tolerance parameter $\gamma$, data $S = ((x_1, y_1), \ldots, (x_m, y_m))$, accesss to sampling oracle $\mathcal{O}$ for $\mu$ and PAC learner $\mathcal{A}_{\mathcal{H}}$.
Initialize $S' = \emptyset$
**for** $i = 1$ to $m$ **do**
    Sample $x_i' \sim \mathcal{B}_{(1+\gamma)r}(x_i)$
    Add $(x_i', y_i)$ to $S'$
**end for**
Set $\hat{h} = \mathcal{A}_{\mathcal{H}}(S')$
**Output:** $\bar{h}_{\gamma r}$ defined by
    $\bar{h}_{\gamma r}(x) = \mathbb{1}\left[\mathbb{E}_{x' \sim \mathcal{B}_{\gamma r}(x)} \hat{h}(x') \geq 1/2\right]$

---

The following is the main result of this section.

**Theorem 10.** *Let $(X, \text{dist})$ be an any metric space with doubling dimension $d$ and doubling measure $\mu$. Let $\mathcal{O}$ be a sampling oracle for $\mu$. Let $\mathcal{H}$ be a hypothesis class and $\mathcal{P}$ a set of distributions over $X \times Y$. Assume $\mathcal{A}_{\mathcal{H}}$ PAC learns $\mathcal{H}$ with $m_{\mathcal{H}}(\epsilon, \delta)$ samples in the realizable setting. Then there exists a learner $\mathcal{A}$, namely TPaS, that*

- *Tolerantly PAC learns $(\mathcal{H}, \mathcal{P}, \text{dist})$ in the tolerantly realizable setting with sample complexity bounded by $m(\epsilon, \delta, \gamma) = O\left(m_{\mathcal{H}}(\epsilon, \delta) \cdot (1 + 1/\gamma)^{\zeta d}\right) = O\left(\frac{\text{VC}(\mathcal{H}) + \log 1/\delta}{\epsilon} \cdot (1 + 1/\gamma)^{\zeta d}\right)$, where $\gamma$ is the tolerance parameter and $d$ is the doubling dimension.*

- *Makes only one query to $\mathcal{A}_{\mathcal{H}}$*

- *Makes $m(\epsilon, \delta, \gamma)$ queries to sampling oracle $\mathcal{O}$*

The proof of this theorem uses the following key technical lemma (proof can be found in Appendix C):

**Lemma 11.** *Let $r > 0$ be a perturbation radius, $\gamma > 0$ a tolerance parameter, and $g : X \to Y$ a classifier. For $x \in X$ and $y \in Y = \{0, 1\}$, we define*

$$\Sigma_{g,y}(x) = \mathbb{E}_{z \sim \mathcal{B}_{r(1+\gamma)}(x)} \mathbb{1}\left[g(z) \neq y\right] \quad and \quad \sigma_{g,y}(x) = \mathbb{E}_{z \sim \mathcal{B}_{r\gamma}(x)} \mathbb{1}\left[g(z) \neq y\right].$$

*Then $\Sigma_{g,y}(x) \leq \frac{1}{3} \cdot \left(\frac{1+\gamma}{\gamma}\right)^{-\zeta d}$ implies that $\sigma_{g,y}(z) \leq 1/3$ for all $z \in \mathcal{B}_r(x)$.*

*Proof of Theorem 10.* Consider some $\epsilon_0 > 0$ and $0 < \delta < 1$ to be given (we will pick a suitable value of $\epsilon_0$ later), and assume the PAC learner $\mathcal{A}_{\mathcal{H}}$ was invoked on the perturbed sample $S'$ of size at least $m_A(\epsilon_0, \delta)$. According to definition 3, this implies that with probability $1 - \delta$, the output $\hat{h} = \mathcal{A}_{\mathcal{H}}(S)$ has (binary) loss at most $\epsilon_0$ with respect to the data-generating distribution. Note that the relevant distribution here is the two-stage process of the original data generating distribution $P$ and the perturbation sampling according to $\mathcal{V} = \mathcal{B}_{(1+\gamma)r}$. Since $P$ is $\mathcal{V}$-robustly realizable, the two-stage process yields a realizable distribution with respect to the standard $0/1$-loss. Thus, we have

$$\mathbb{E}_{(x,y) \sim P} \mathbb{E}_{z \sim \mathcal{B}_{r(1+\gamma)}(x)} \mathbb{1}\left[\hat{h}(z) \neq y\right] \leq \epsilon_0.$$

With Lemma 11, this becomes $\mathbb{E}_{(x,y) \sim P} \Sigma_{\hat{h},y}(x) \leq \epsilon_0$. For $\lambda > 0$, Markov's inequality then yields :

$$\mathbb{E}_{(x,y) \sim P} \mathbb{1}\left[\Sigma_{\hat{h},y}(x) \leq \lambda\right] > 1 - \epsilon_0/\lambda \tag{1}$$

Thus setting $\lambda = \frac{1}{3} \cdot \left(\frac{1+\gamma}{\gamma}\right)^{-\zeta d}$ and plugging in the result of the Lemma 11 to equation (1), we get

$$\mathbb{E}_{(x,y) \sim P} \mathbb{1}\left[\forall z \in \mathcal{B}_r(x), \sigma_{\hat{h},y}(z) \leq 1/3\right] > 1 - \epsilon_0/\lambda.$$

Since $\sigma_{\hat{h},y}(z) \leq 1/3$ implies that $\mathbb{1}\left[\mathbb{E}_{z' \sim \mathcal{B}_{\gamma r}(z)} \hat{h}(z') \geq 1/2\right] = y$, using the definition of the smoothed classifier $\bar{h}_{\gamma r}$ we get

$$\mathbb{E}_{(x,y) \sim P} \mathbb{1}\left[\exists z \in \mathcal{B}_r(x), \bar{h}_{\gamma r}(z) \neq y\right] \leq \epsilon_0/\lambda, \tag{2}$$

which implies $\mathcal{L}_P^r(\bar{h}_{\gamma r}) \leq \epsilon_0/\lambda$. Thus, for the robust learning problem, if we are given a desired accuracy $\epsilon$ and we want $\mathcal{L}_P^r(\bar{h}_{\gamma r}) \leq \epsilon$, we can pick $\epsilon_0 = \lambda \epsilon$. Putting it all together, we get sample complexity $m \leq O(\frac{\text{VC}(\mathcal{H}) + \log 1/\delta}{\epsilon_0})$ where $\epsilon_0 = \lambda \epsilon$, and $\lambda = \frac{1}{3} \cdot \left(\frac{1+\gamma}{\gamma}\right)^{-\zeta d}$. Therefore,

$$m \leq O\left(\frac{\text{VC}(\mathcal{H}) + \log 1/\delta}{\epsilon} \cdot (1 + 1/\gamma)^{\zeta d}\right). \qquad \square$$

Since the dependence on $d$ is exponential, the algorithm becomes impractical for high dimensions if $\gamma$ is very small. However, since $\gamma$ represents our uncertainty in the value of the true perturbation radius, it is natural to assume that it is a small but positive number. We can therefore ask whether there exists a threshold for each dimension such that if $\gamma$ is above the threshold we can learn efficiently. In

particular, for any constant $c > 0$, we can ensure that $(1 + 1/\gamma)^{\zeta d} \leq 1 + c$ if we set $\gamma \geq \frac{\zeta d}{c}$. Thus the sample complexity of our learner does not depend on the dimension as long as $\gamma \geq \frac{\zeta d}{c}$. For example, for a Euclidean space with the $\ell_\infty$ metric, we have $\zeta = 1$. Therefore setting $c = 1000$ would let us use a small $\gamma$ for dimensions up to 1000.

**Computational complexity of the learner.** Assuming we have access to $\mathcal{O}$ and an efficient algorithm for non-robust PAC-learning in the realizable setting, we can compute $\hat{h}$ efficiently. Therefore, the learning can be done efficiently in this case. However, at the prediction time, we need to compute $\bar{h}(x)$ on new test points which requires us to compute an expectation. We can instead *estimate* the expectations using random samples from the sampling oracle. For a single test point $x$, if the number of samples we draw is $\Omega(\log 1/\delta)$ then with probability at least $1 - \delta$ we get the same result as that of the optimal $\bar{h}(x)$. Using more samples we can boost this probability to guarantee a similar output to that of $\bar{h}$ on a larger set of test points.

# 6 Improved tolerant learning guarantees through sample compression

The perturb-and-smooth approach discussed in the previous section offers a general method for tolerant robust learning. However, one shortcoming of this approach is the exponential dependence of its sample complexity with respect to the doubling dimension of the metric space. Furthermore, the tolerant robust guarantee relied on the data generating distribution being tolerantly realizable. In this section, we propose another approach that addresses both of these issues. The idea is to adopt the perturb-and-smooth approach within a sample compression argument. We introduce the notion of a $(\mathcal{U}, \mathcal{V})$-tolerant sample compression scheme and present a learning bound based on such a compression scheme, starting with the realizable case. We then show that this implies learnability in the agnostic case as well. Remarkably, this tolerant compression based analysis will yield bounds on the sample complexity that avoid the exponential dependence on the doubling dimension.

For a compact representation, we will use the general notation $\mathcal{U}, \mathcal{V}$, and $\mathcal{W}$ for the three perturbation types (actual, reference and smoothing type) in this section and will assume that they satisfy the Property 1 below for some parameter $\beta > 0$. Lemma 11 implies that, in the metric setting, for any radius $r$ and tolerance parameter $\gamma$ the perturbation types $\mathcal{U} = \mathcal{B}_r$, $\mathcal{V} = \mathcal{B}_{(1+\gamma)r}$, and $\mathcal{W} = \mathcal{B}_{\gamma r}$ have this property for $\beta = \frac{1}{3} \left( \frac{1+\gamma}{\gamma} \right)^{-\zeta d}$.

**Property 1.** *For a fixed $0 < \beta < 1/2$, we assume that the perturbation types $\mathcal{V}, \mathcal{U}$ and $\mathcal{W}$ are so that for any classifier $h$ and any $x \in X$, any $y \in \{0, 1\}$ if*

$$\mathbb{E}_{z \sim \mathcal{V}(x)}[h(z) = y] \geq 1 - \beta$$

*then $\mathcal{W}$-smoothed class classifier $\bar{h}_{\mathcal{W}}$ satisfies $\bar{h}_{\mathcal{W}}(z) = y$ for all $z \in \mathcal{U}(x)$.*

A compression scheme of size $k$ is a pair of functions $(\kappa, \rho)$, where the compression function $\kappa : \bigcup_{i=1}^{\infty} (X \times Y)^i \to \bigcup_{i=1}^{k} (X \times Y)^i$ maps samples $S = ((x_1, y_1), (x_2, y_2), \ldots, (x_m, y_m))$ of arbitrary size to sub-samples of $S$ of size at most $k$, and $\rho : \bigcup_{i=1}^{k} (X \times Y)^i \to Y^X$ is a decompression function that maps samples to classifiers. The pair $(\kappa, \rho)$ is a sample compression scheme for loss $\ell$ and class $\mathcal{H}$, if for any samples $S$ realizable by $\mathcal{H}$, we recover the correct labels for all $(x, y) \in S$, that is, $\mathcal{L}_S(H) = 0$ implies that $\mathcal{L}_S(\kappa \circ \rho(S)) = 0$.

For tolerant learning, we introduce the following generalization of compression schemes:

**Definition 12** (Tolerant sample compression scheme). *A sample compression scheme $(\kappa, \rho)$ is a $\mathcal{U}, \mathcal{V}$-tolerant sample compression scheme for class $\mathcal{H}$, if for any samples $S$ that are $\ell^{\mathcal{V}}$ realizable by $\mathcal{H}$, that is $\mathcal{L}_S^{\mathcal{V}}(\mathcal{H}) = 0$, we have $\mathcal{L}_S^{\mathcal{U}}(\kappa \circ \rho(S)) = 0$.*

The next lemma establishes that the existence of a sufficiently small tolerant compression scheme for the class $\mathcal{H}$ yields bounds on the sample complexity of tolerantly learning $\mathcal{H}$. The proof of the lemma is based on a modification of a standard compression based generalization bound. Appendix Section D provides more details.

**Lemma 13.** *Let $\mathcal{H}$ be a hypothesis class and $\mathcal{U}$ and $\mathcal{V}$ be perturbation types with $\mathcal{U}$ included in $\mathcal{V}$. If the class $\mathcal{H}$ admits a $(\mathcal{U}, \mathcal{V})$-tolerant compression scheme of size bounded by $k \ln(m)$ for sample of size $m$, then the class is $(\mathcal{U}, \mathcal{V})$-tolerantly learnable in the realizable case with sample complexity bounded by $m(\epsilon, \delta) = \tilde{O} \left( \frac{k + \ln(1/\delta)}{\epsilon} \right)$.*

347  We next establish a bound on the tolerant compression size for general VC-classes, which will then
348  immediately yield the improved sample complexity bounds for tolerant learning in the realizable case.
349  The proof is sketched here; its full version has been moved to the Appendix for lack of space.

350  **Lemma 14.** *Let $\mathcal{H} \subseteq Y^X$ be some hypothesis class with finite VC-dimension $\mathrm{VC}(\mathcal{H}) < \infty$, and*
351  *let $\mathcal{U}, \mathcal{V}, \mathcal{W}$ satisfy the conditions in Property 1 for some $\beta > 0$. Then there exists a $(\mathcal{U}, \mathcal{V})$-tolerant*
352  *sample compression scheme for $\mathcal{H}$ of size $\tilde{O}\left(\mathrm{VC}(\mathcal{H}) \ln(\frac{m}{\beta})\right)$.*

353  *Proof Sketch.* We will employ a boosting-based approach to establish the claimed compression sizes.
354  Let $S = ((x_1, y_1), (x_2, y_2), \ldots, (x_m, y_m))$ be a data-set that is $\ell^{\mathcal{V}}$-realizable with respect to $\mathcal{H}$. We
355  let $S_{\mathcal{V}}$ denote an "inflated data-set" that contains all domain points in the $\mathcal{V}$-perturbation sets of
356  the $x_i \in S^X$, that is $S_{\mathcal{V}}^X := \bigcup_{i=1}^m \mathcal{V}(x_i)$. Every point $z \in S_{\mathcal{V}}^X$ is assigned the label $y = y_i$ of the
357  minimally-indexed $(x_i, y_i) \in S$ with $z \in \mathcal{V}(x_i)$, and we set $S_{\mathcal{V}}$ to be the resulting collection of
358  labeled data-points.

359  We then use the boost-by-majority method to encode a classifier $g$ that (roughly speaking) has error
360  bounded by $\beta/m$ over (a suitable measure over) $S_{\mathcal{V}}$. This boosting method outputs a $T$-majority
361  vote $g(x) = \mathbb{1}\left[\Sigma_{i=1}^T h_i(x)\right] \geq 1/2$ over weak learners $h_i$, which in our case will be hypotheses from
362  $\mathcal{H}$. We prove that this error can be achieved with $T = 18 \ln(\frac{2m}{\beta})$ rounds of boosting. We prove that
363  each weak learner that is used in the boosting procedure can be encoded with $n = \tilde{O}(\mathrm{VC}(\mathcal{H}))$ many
364  sample points from $S$. The resulting compression size is thus $n \cdot T = \tilde{O}\left(\mathrm{VC}(\mathcal{H}) \ln(\frac{m}{\beta})\right)$.

365  Finally, the error bound $\beta/m$ of $g$ over $S_{\mathcal{V}}$ implies that the error in each perturbation set $\mathcal{V}(x_i)$ of a
366  sample point $(x_i, y_i) \in S$ is at most $\beta$. Property 1 then implies $\mathcal{L}_S^{\mathcal{U}}(\bar{g}_{\mathcal{W}}) = 0$ for the $\mathcal{W}$-smoothed
367  classifier $\bar{g}_{\mathcal{W}}$, establishing the $(\mathcal{U}, \mathcal{V})$-tolerant correctness of the compression scheme. $\qquad\square$

368  This yields the following result

369  **Theorem 15.** *Let $\mathcal{H}$ be a hypothesis class of finite VC-dimension and $\mathcal{V}, \mathcal{U}, \mathcal{W}$ be three perturbation*
370  *types (actual, reference and smoothing) satisfying Property 1 for some $\beta > 0$. Then the sample*
371  *complexity (omitting log-factors) of $(\mathcal{U}, \mathcal{V})$-tolerantly learning $\mathcal{H}$ is bounded by*

$$m(\epsilon, \delta) = \tilde{O}\left(\frac{\mathrm{VC}(\mathcal{H}) \ln(1/\beta) + \ln(1/\delta)}{\epsilon}\right)$$

372  *in the realizable case, and in the agnostic case by*

$$m(\epsilon, \delta) = \tilde{O}\left(\frac{\mathrm{VC}(\mathcal{H}) \ln(1/\beta) + \ln(1/\delta)}{\epsilon^2}\right)$$

373  *Proof.* The bound for the realizable case follows immediately from Lemma 14 and the subsequent
374  discussion (in the Appendix). For the agnostic case, we employ a reduction from agnostic robust
375  learnabilty to realizable robust learnability (Montasser et al., 2019; Moran & Yehudayoff, 2016).
376  The reduction is analogous to the one presented in Appendix C of Montasser et al. (2019) for usual
377  (non-tolerant) robust learnablity with some minor modifications. Namely, for a sample $S$, we choose
378  the largest subsample $S'$ that is $\ell^{\mathcal{V}}$-realizable (this will result in competitiveness with a $\ell^{\mathcal{V}}$-optimal
379  classifier), and we will use the boosting procedure described there for the $\ell^{\mathcal{U}}$ loss. For the sample sizes
380  employed for the weak learners in that procedure, we can use the sample complexity for $\epsilon = \delta = 1/3$
381  of an optimal $(\mathcal{U}, \mathcal{V})$-tolerant learner in the realizable case (note that each learning problem during
382  the boosting procedure is a realizable $(\mathcal{U}, \mathcal{V})$-tolerant learning task). These modifications result in the
383  stated sample complexity for agnostic tolerant learnability. $\qquad\square$

384  In particular, for the doubling measure scenario (as considered in the previous section), we obtain

385  **Corollary 16.** *For metric tolerant learning with tolerance parameter $\gamma$ in doubling di-*
386  *mension $d$ the sample complexity of learning in the realizable case is bounded by*
387  $m(\epsilon, \delta) = \tilde{O}\left(\frac{\mathrm{VC}(\mathcal{H})\zeta d \ln(1+1/\gamma) + \ln(1/\delta)}{\epsilon}\right)$ *and in the agnostic case by* $m(\epsilon, \delta) =$
388  $\tilde{O}\left(\frac{\mathrm{VC}(\mathcal{H})\zeta d \ln(1+1/\gamma) + \ln(1/\delta)}{\epsilon^2}\right)$.

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
