## A Standard results from VC theory

Let $X$ be a domain. For hypothesis $h$ and $B \subseteq X$ let $h(B) = (h(b))_{b \in B}$.

**Definition 17** (VC-dimension). *We say $\mathcal{H}$ shatters $B \subseteq X$ if $|\{h(B) : h \in \mathcal{H}\}| = 2^{|B|}$. The VC-dimension of $\mathcal{H}$, denoted by $\mathrm{VC}(\mathcal{H})$, is defined to be the supremum of the size of the sets that are shattered by $\mathcal{H}$.*

**Theorem 18** (Existence of Realizable PAC Learners Hanneke (2016); Simon (2015); Blumer et al. (1989)). *Let $\mathcal{H}$ be a hypothesis class with bounded VC dimension. Then $\mathcal{H}$ is PAC learnable in the realizable setting using $O\left(\frac{\mathrm{VC}(\mathcal{H}) + \log(1/\delta)}{\epsilon}\right)$ samples.*

**Theorem 19** (Existence of Agnostic PAC Learners Haussler (1992)). *Let $\mathcal{H}$ be a hypothesis class with bounded VC dimension. Then $\mathcal{H}$ is PAC learnable in the agnostic setting using $O\left(\frac{\mathrm{VC}(\mathcal{H}) + \log(1/\delta)}{\epsilon^2}\right)$ samples.*

## B Metric spaces

**Definition 20.** *A metric space $(X, \mathrm{dist})$ is called a doubling metric if there exists a constant $M$ such that every ball of radius $r$ in it can be covered by at most $M$ balls of radius $r/2$. The quantity $\log_2 M$ is called the doubling dimension.*

**Definition 21.** *For a metric space $(X, \mathrm{dist})$, a measure $\mu$ defined on $X$ is called a doubling measure if there exists a constant $C$, such that for all $x \in X$ and $r \in \mathbb{R}^+$, we have that $0 < \mu(\mathcal{B}_{2r}(x)) \leq C \cdot \mu(\mathcal{B}_r(x)) < \infty$. In this case, $\mu$ is called $C$-doubling.*

It can be shown (Luukkainen & Saksman, 1998) that every *complete* metric with doubling dimension $d$ has a $C$-doubling measure $\mu$ for some $C \leq 2^{cd}$ where $c$ is a universal constant. For example, Euclidean spaces with an $\ell_p$ distance metric are complete and the Lesbesgue measure is a doubling measure.

The following lemmas follow straightforwardly from the definitions doubling metric and measures.

**Lemma 22.** *Let $(X, \mathrm{dist})$ be a doubling metric equipped with a $C$-doubling measure $\mu$. Then for all $x \in X$, $r > 0$, and $\alpha > 1$, we have that $\mu(\mathcal{B}_{\alpha r}(x)) \leq C^{\lceil \log_2 \alpha \rceil} \cdot \mu(\mathcal{B}_r(x))$*

*Proof.* Since $\mu$ is a measure, if $B, B' \subseteq X$ such that $B \subseteq B'$, then $\mu(B) \leq \mu(B')$. Let $R = 2^{\lceil \log_2 \alpha \rceil}$. It's clear that $R \geq \alpha$. Therefore $\mathcal{B}_{\alpha r}(x) \subseteq \mathcal{B}_R(x)$. Expanding $\mathcal{B}_r(x)$ by a factor of two $\lceil \log_2 \alpha \rceil$ times, we get $\mathcal{B}_R(x)$, which means $\mu(\mathcal{B}_R(x)) \leq C^{\lceil \log_2 \alpha \rceil} \cdot \mu(\mathcal{B}_r(x))$. But since $\mathcal{B}_{\alpha r}(x) \subseteq \mathcal{B}_R(x)$, we get the desired result. $\square$

**Lemma 23.** *Let $(X, \mathrm{dist})$ be a doubling metric equipped with a $C$-doubling measure $\mu$. Let $x, x' \in X$, $r > 0$, and $\alpha > 1$ be such that $\mathcal{B}_r(x') \subseteq \mathcal{B}_{\alpha r}(x)$. Then $\mu(\mathcal{B}_{\alpha r}(x)) \leq C^{\lceil \log_2(2\alpha) \rceil} \cdot \mu(\mathcal{B}_r(x'))$.*

*Proof.* By Lemma 22, all we need to show is that $B_{\alpha r}(x) \subseteq \mathcal{B}_{2\alpha r}(x')$. Indeed, let $y \in \mathcal{B}_{\alpha r}(x)$ be any point. Then, from triangle inequality, we have that

$$d(x', y) \leq d(x, x') + d(x, y)$$
$$\leq d(x, x') + \alpha r$$

Moreover, since $x' \in \mathcal{B}_{\alpha r}(x)$, we have that $d(x, x') \leq \alpha r$. Substituting into the equation above, we get $d(x', y) \leq 2\alpha r$, which means $y \in \mathcal{B}_{2\alpha r}(x')$. $\square$

Finally, we also get:

**Lemma 24.** *For any family $\mathcal{M}$ of complete, doubling metric spaces, there exist constants $c_1, c_2 > 0$ such that for any metric space $(X, \mathrm{dist}) \in \mathcal{M}$ with doubling dimension $d$, there exists a measure $\mu$ such that if a ball $\mathcal{B}_r$ of radius $r > 0$ is completely contained inside a ball $\mathcal{B}_{\alpha r}$ of radius $\alpha r$ (with potentially a different center) for any $\alpha > 1$, then $0 < \mu(\mathcal{B}_{\alpha r}) \leq (c_1 \alpha)^{c_2 d} \mu(\mathcal{B}_r)$.*

*Proof.* We prove this when $\mathcal{M}$ is the set of all complete, doubling metric spaces employing Lemmas 22 and 23, that can be found in Appendix, part B. We have that $C^{\lceil \log_2(2\alpha) \rceil} \leq (2\alpha)^{2\log_2 C}$. Since $\log_2 C \leq cd$, we get $(2\alpha)^{2\log_2 C} \leq (2\alpha)^{cd}$. Thus $c_1 = 2$ and $c_2 = c$. $\qquad \square$

**Corollary 25.** *Suppose we have a constant $\alpha_0 > 1$ such that we know that $\alpha \geq \alpha_0$. Then the bound in Lemma 24 can be further simplified to $0 < \mu(\mathcal{B}_{\alpha r}) \leq \alpha^{\zeta d}\mu(\mathcal{B}_r)$, where $\zeta$ depends on $\mathcal{M}$ and $\alpha_0$. Furthermore, if $c_1 = 1$ then we can set $\alpha_0 = 1$.*

*Proof.* $(c_1\alpha)^{c_2 d} = \alpha^{c_2 d(1+\log_\alpha c_1)} \leq \alpha^{c_2 d(1+\log_{\alpha_0} c_1)} = \alpha^{\zeta d}$ for $\zeta = c_2(1 + \log_{\alpha_0} c_1)$. If $c_1 = 1$, then $\zeta = c_2$ for all $\alpha$. $\qquad \square$

# C  Proof of Lemma 11

Let $X_{\mathrm{err}} = \{z \in \mathcal{B}_{r(1+\gamma)}(x) \mid g(z) \neq y\}$. Then, we have that $\Sigma_{g,y}(x) = \mathbb{E}_{z \sim \mathcal{B}_{r(1+\gamma)}(x)} \mathbb{1}[g(z) \neq y] = \frac{\mu(X_{\mathrm{err}})}{\mu(\mathcal{B}_{r(1+\gamma)}(x))}$. Further, for all $z \in \mathcal{B}_r(x)$, we have $\mathbb{E}_{z' \sim \mathcal{B}_{r\gamma}(z)} \mathbb{1}[g(z') \neq y] = \frac{\mu(X_{\mathrm{err}} \cap \mathcal{B}_{r\gamma}(z))}{\mu(\mathcal{B}_{r\gamma}(z))}$.

Let $z \in \mathcal{B}_r(x)$. Since this implies that $\mathcal{B}_{r\gamma}(z) \subseteq \mathcal{B}_{r(1+\gamma)}(x)$, the worst case happens when $X_{\mathrm{err}} \subseteq \mathcal{B}_{r\gamma}(z)$. Therefore,

$$
\begin{aligned}
\sigma_{g,y}(x) &= \mathbb{E}_{z' \sim \mathcal{B}_{r\gamma}(z)} \mathbb{1}[g(z') \neq y] \qquad\qquad (3)\\
&= \frac{\mu(X_{\mathrm{err}} \cap \mathcal{B}_{r\gamma}(z))}{\mu(\mathcal{B}_{r\gamma}(z))}\\
&\leq \frac{\mu(X_{\mathrm{err}})}{\mu(\mathcal{B}_{r\gamma}(z))}\\
&\leq \frac{\Sigma_{g,y}(x) \cdot \mu(\mathcal{B}_{r(1+\gamma)}(x))}{\mu(\mathcal{B}_{r\gamma}(z))}\\
&\leq \Sigma_{g,y}(x) \cdot \left(\frac{1+\gamma}{\gamma}\right)^{\zeta d},
\end{aligned}
$$

where the last inequality is implied by Lemma 24. Thus, $\Sigma(x) \leq \frac{1}{3} \cdot \left(\frac{1+\gamma}{\gamma}\right)^{-\zeta d}$ implies that $\sigma(z) \leq 1/3$ as claimed.

# D  Compression based bounds

## D.1  Proof of Lemma 13

To prove the generalization bound for tolerant learning, we employ the following lemma that establishes generalization for compression schemes for adversarial losses:

**Lemma 26** (Lemma 11, (Montasser et al., 2019)). *For any $k \in \mathbb{N}$ and fixed function $\rho : \bigcup_{i=1}^{k}(X \times Y)^i \to Y^X$, for any distribution $P$ over $X \times Y$ and any $m \in \mathbb{N}$, with probability at least $(1 - \delta)$ over an i.i.d. sample $S = ((x_1, y_1), (x_2, y_2), \dots, (x_m, y_m))$: if there exist indices $i_1, i_2, \dots, i_k$ such that*

$$
\mathcal{L}_S^{\mathcal{U}}(\rho((x_{i_1}, y_{i_1}), (x_{i_2}, y_{i_2}), \dots, (x_{i_k}, y_{i_k}))) = 0
$$

*then the robust loss of the decompression with respect to $P$ is bounded by*

$$
\mathcal{L}_P^{\mathcal{U}}(\rho((x_{i_1}, y_{i_1}), (x_{i_2}, y_{i_2}), \dots, (x_{i_k}, y_{i_k})))
$$
$$
\leq \frac{1}{m-k}(k\ln(m) + \ln(1/\delta))
$$

The above lemma implies that if $(\kappa, \rho)$ is a compression scheme that compresses data-sets of size $m$ to at most $k\ln(m)$ data points, for class $\mathcal{H}$ and robust loss $\ell^{\mathcal{U}}$, then the sample complexity (omitting logarithmic factors) of robustly learning $\mathcal{H}$ in the realizable case is bounded by

$$
m(\epsilon, \delta) = \tilde{O}\left(\frac{k + \ln(1/\delta)}{\epsilon}\right)
$$

For the tolerant setting, since every sample that is realizable with respect to $\ell^{\mathcal{V}}$ is also realizable with respect to $\ell^{\mathcal{U}}$, if a $(\mathcal{U}, \mathcal{V})$-tolerant compression scheme compresses to at most $k \ln(m)$ data-points and decompresses all $\ell^{\mathcal{V}}$-realizable samples $S$ to functions that have $\ell^{\mathcal{U}}$-loss 0 on $S$, then the lemma implies the above bound for the $(\mathcal{U}, \mathcal{V})$-tolerant sample complexity of learning $\mathcal{H}$.

## D.2 Proof of Lemma 14

The proof of this Lemma employs the notions of a sample being $\epsilon$-net or and $\epsilon$-approximation for a hypothesis class $\mathcal{H}$. A labeled data set $S = ((x_1, y_1), (x_2, y_2), \ldots, (x_m, y_m))$ is an $\epsilon$-net for class $\mathcal{H}$ with respect to distribution $P$ over $X \times Y$ if for every hypothesis $h \in \mathcal{H}$ with $\mathcal{L}_P^{0/1}(h) \geq \epsilon$, there exists an index $j$ and $(x_j, y_j) \in S$ with $h(x_j) \neq y_j$. $S$ is an $\epsilon$-approximation for class $\mathcal{H}$ with respect to distribution $P$ over $X \times Y$ if for every hypothesis $h \in \mathcal{H}$ we have $|\mathcal{L}_S^{0/1}(h) - \mathcal{L}_P^{0/1}(h)| \leq \epsilon$. Standard VC-theory tells us that, for classes with bounded VC-dimension, sufficiently large samples from $P$ are $\epsilon$-nets or $\epsilon$-approximations with high probability (Haussler & Welzl, 1987).

*Proof.* We will employ a boosting-based approach to establish the claimed compression sizes. Let $S = ((x_1, y_1), (x_2, y_2), \ldots, (x_m, y_m))$ be a data-set that is $\ell^{\mathcal{V}}$-realizable with respect to $\mathcal{H}$. We let $S_{\mathcal{V}}$ denote an "inflated data-set" that contains all domain points in the perturbation sets of the $x_i \in S^X$, that is

$$S_{\mathcal{V}}^X := \bigcup_{i=1}^{m} \mathcal{V}(x_i)$$

Every point $z \in S_{\mathcal{V}}^X$ is assigned the label $y = y_i$ of the minimally-indexed $(x_i, y_i) \in S$ with $z \in \mathcal{V}(x_i)$, and we set $S_{\mathcal{V}}$ to be the resulting collection of labeled data-points. (Note that since the sample $S$ is assumed to be $\ell^{\mathcal{V}}$-realizable, assigning it the label of some other corresponding data point in case $z \in \mathcal{V}(x_i) \cap \mathcal{V}(x_j)$ for $x_i \neq x_j$, would not induce any inconsistencies). Now let $D$ be the probability measure over $S_{\mathcal{V}}^X$ defined by first sampling an index $j$ uniformly from $[j] = \{1, 2, \ldots, j\}$ and then sampling a domain point $z \sim \mathcal{V}(x_j)$ from the $\mathcal{V}$-perturbation set around the $j$-th sample point in $S$. Note that this implies that if $D(B) \leq (\beta/m)$ for some subset $B \subseteq S_{\mathcal{V}}^X$, then

$$\mathbb{P}_{z \sim \mathcal{V}(x)}[z \in B] \leq \beta \tag{4}$$

for all $x \in S^X$.

We will now show that, by means of a compression scheme, we can encode a hypothesis $g$ with binary loss

$$\mathcal{L}_D^{0/1}(g) \leq \beta/m. \tag{5}$$

Property 1 together with Equation 4 then implies that the resulting $\mathcal{W}$-smoothed function $\bar{g}$ has $\mathcal{U}$-robust loss 0 on the sample $S$, $\mathcal{L}_S^{\mathcal{U}}(\bar{g}) = 0$. Since the smoothing is a deterministic operation once $g$ is fixed, this implies the existence of a $(\mathcal{U}, \mathcal{V})$-tolerant compression scheme.

Standard VC-theory tells us that, for a class $G$ of bounded VC-dimension, for any distribution over $X \times Y$, and any $\epsilon, \delta > 0$, with probability at least $(1 - \delta)$ an i.i.d. sample of size $\Theta\left(\frac{\text{VC}(G) + \ln(1/\delta)}{\epsilon^2}\right)$ is an $\epsilon$-approximation for the class $G$ (Haussler & Welzl, 1987). This implies in particular, that there exists a finite subset $S_{\mathcal{V}}^f \subset S_{\mathcal{V}}$ of size at most $\frac{4m^2 C \cdot \text{VC}(G)}{\beta^2}$ (for some constant $C$) with the property that any classifier $g \in G$ with empirical (binary) loss at most $\beta/2m$ on $S_{\mathcal{V}}^f$ has loss $\mathcal{L}_D^{0/1}(g) \leq \beta/m$ with respect to the distribution $D$. We will choose such a set $S_{\mathcal{V}}^f$ for the class $G$ of $T$-majority votes over $\mathcal{H}$ for $T = 18 \ln(\frac{2m}{\beta})$. That is

$$G = \{g \in Y^X \mid \exists h_1, h_2, \ldots, h_T \in H :$$
$$g(x) = \mathbb{1}\left[\Sigma_{i=1}^T h_i(x) \geq 1/2\right]\}$$

The VC-dimension of $G$ is bounded by $\text{VC}(G) = \mathcal{O}(T \cdot \text{VC}(\mathcal{H}) \log(T\text{VC}(\mathcal{H}))) = \mathcal{O}(18 \ln(\frac{2m}{\beta})\text{VC}(\mathcal{H}) \log(18 \ln(\frac{2m}{\beta})\text{VC}(\mathcal{H})))$ (Shalev-Shwartz & Ben-David, 2014).

We will now show how to obtain the classifier $g$ by means of a boosting approach on the finite data-set $S_{\mathcal{V}}^f$. More specifically, we will use the boost-by-majority method. This method outputs a $T$-majority

vote $g(x) = \mathbb{1}\left[\Sigma_{i=1}^{T} h_i(x)\right] \geq 1/2$ over weak learners $h_i$, which in our case will be hypotheses from $\mathcal{H}$. After $T$ iterations with $\gamma$-weak learners, the empirical loss over the sample $S_{\mathcal{V}}^{f}$ is bounded by $e^{-2\gamma^2 T}$ (see Section 13.1 in (Schapire & Freund, 2013)). Thus, with $\gamma = 1/6$, and $T = 18\ln(\frac{2m}{\beta})$, we obtain

$$\mathcal{L}_{S_{\mathcal{V}}^{f}}^{0/1}(g) \leq \frac{\beta}{2m}$$

which, by the choice of $S_{\mathcal{V}}^{f}$ implies

$$\mathcal{L}_{D}^{0/1}(g) \leq \beta/m$$

which is what we needed to show according to Equation 5.

It remains to argue that the weak learners to be employed in the boosting procedure can be encoded by a small number of sample points from the original sample $S$. For this part, we will employ a technique introduced earlier for robust compression (Montasser et al., 2019). Recall that the set $S$ is $\mathcal{V}$-robustly realizable, which implies that the set $S_{\mathcal{V}}^{f}$ is (binary loss-) realizable by $\mathcal{H}$. By standard VC-theory, for every distribution $D_i$ over $S_{\mathcal{V}}^{f}$, there exists an $\epsilon$-net of size $\mathcal{O}(\text{VC}(\mathcal{H})/\epsilon)$ (Haussler & Welzl, 1987). Thus, for every distribution $D_i$ over $S_{\mathcal{V}}^{f}$ (that may occur during the boosting procedure), there exists a subsample $S_i$ of $S_{\mathcal{V}}^{f}$, of size at most $n = \mathcal{O}(3\text{VC}(\mathcal{H}))$ with the property that every hypothesis from $\mathcal{H}$ that is consistent with $S_i$ has binary loss at most $1/3$ with respect to $D_i$ (thus can serve as a weak learner for margin $\gamma = 1/6$ in the above procedure). Now for every labeled point $(x, y) \in S_i$, there is a sample point $(x_j, y_j) \in S$ in the original sample $S$ such that $x \in \mathcal{V}(x_j)$ and $y = y_j$. Let $S_i'$ be the collection of these corresponding original sample points. Note that any hypothesis $h \in H$ that is $\mathcal{V}$-robustly consistent with $S_i'$ is consistent with $S_i$. Therefore we can use the $n$ original data-points in $S_i'$ to encode the weak learner $h_i$ (for the decoding any $\mathcal{V}$-robust ERM hypothesis can be chosen to obtain $h_i$).

To summarize, we will compress the sample $S$ to the sequence $S_1', S_2', \ldots S_T'$ of $n \cdot T = \mathcal{O}(\text{VC}(\mathcal{H}) \ln(\frac{m}{\beta}))$ sample points from $S$. To decode, we obtain the function $g$ as a majority vote over the weak learner $h_i$ and proceed to obtain the $\mathcal{W}$-smoothed function $\bar{g}$. This function $\bar{g}$ satisfies $\mathcal{L}_{S}^{\mathcal{U}}(\bar{g}) = 0$ and by this we have established the existence of a $\mathcal{U}, \mathcal{V}$-tolerant compression scheme of size $\mathcal{O}(\text{VC}(\mathcal{H}) \ln(\frac{m}{\beta}))$ as claimed. $\qquad\square$