# OpenReview forum: "Adversarially Robust Learning with Tolerance"
_NeurIPS.cc/2022/Conference — NeurIPS 2022 Submitted_

### Official Review · Reviewer_FjcX · 2022-07-10

**Rating:** 7
**Confidence:** 4
**Soundness:** 3 good
**Presentation:** 4 excellent
**Contribution:** 3 good

**Summary:**

Setting and contributions:
The authors suggest a new realistic setting of adversarially robust PAC learning (to test time attacks) in metric spaces,
such that the learner is tested at inference time on a slightly bigger perturbation set that is controlled by a parameter (gamma>0) for his choice. We refer to it as the tolerance parameter.
Obviously, by choosing a smaller gamma, the perturbation set at test time is getting closer to the one on training time, which is a harder benchmark. As a result, there is a trade-off between sample complexity and competing with a better benchmark.

The main results are two algorithms.
The first one is in the robust realizable setting, with sample complexity that depends exponentially on the metric space dimension (doubling dimension) and linearly in the VC dimension. The algorithm is simple and efficient.

The second algorithm, for the agnostic setting, incorporates the idea of the first one (perturb and smooth) with a sample compression scheme argument, suitable for the model with tolerance. The sample complexity is improved exponentially (linear in the doubling dimension), due to the strong generalization guarantees of sample compression.

**Questions:**

-What is the support of the distribution? Is it the clean examples, and for sampling from a perturbation set you first sample from a clean example?
More specifically, is it defined similarly to the previous papers on theoretical robust learning?

-Can you please give a short explanation of how you avoid the dual VC in Lemma 14? I know why it pops up in the case of arbitrary perturbation sets.
I read the proof, but not sure about the answer.

I would be happy to figure it out and recommend the paper for acceptance.

**Ethics Review Area:**

["I don’t know"]

**Limitations:**

-

**Strengths And Weaknesses:**

This theoretical setting is natural, and well-motivated by empirical work on adversarial examples.

The results are satisfying, specifically exploiting the metric space assumption for removing the dual VC that pops up in the case of general perturbation sets.

I find the techniques simple and elegant.

Overall, I think that it is a nice idea for studying robustness.

---

> ### Author Response · Authors · 2022-08-02
> **Please see the response below**
>
> Thanks for the detailed review!
>
> >What is the support of the distribution? Is it the clean examples, and for sampling from a perturbation set you first sample from a clean example? More specifically, is it defined similarly to the previous papers on theoretical robust learning?
>
> Yes, it is defined similarly to previous work. The data distribution $P$ is where the clean examples are generated from. The space of perturbed examples can, however, be bigger than the support of $P$. The distribution we use for the perturb-and-smooth approach is the uniform distribution over perturbation sets convolved with $P$, i.e., we indeed sample from $P$ and then for each sampled point, we sample again (uniformly) from the perturbation set centered on that point.
>
> >Can you please give a short explanation of how you avoid the dual VC in Lemma 14? I know why it pops up in the case of arbitrary perturbation sets. I read the proof, but not sure about the answer.
>
> At a high level, it is possible to avoid the dependence on dual-VC because of the metric structure of the domain set and the tolerant framework: if we can encode a classifier with small error (exponentially small with respect to the doubling dimension) on the perturbed distribution w.r.t. larger perturbation sets, then we can use smoothing to get a classifier that correctly classifies “every point” in the inner inflated sets This is, however, not the case for spaces with infinite doubling dimension and a smoothing step does not necessarily help without tolerance. For example, Montasser et al.’s compression scheme does not use boosting+smoothing (because of the non-tolerant setting and the absence of this metric structure).
>
> In more details, we define a tolerant compression scheme (Definition 12) that borrows ideas from our perturb-and-smooth algorithm. More specifically, we define the perturbed distribution over the sample that we want to compress, and then use boosting to build a classifier with very small error with respect to this distribution. The nice property of boosting is that its error decreases exponentially with the number of iterations, as a result we also get linear dependence on the doubling dimension. This classifier can be encoded using $O(T.VC)$ samples ($T$ rounds of boosting, and each weak classifier can be encoded using $O(VC)$ samples). Our decoder receives the description of these weak classifiers, combines them, and performs a final smoothing step. The smoothing step translates the exponentially small error with respect to the perturbed distribution to zero error with respect to the (inner) inflated set. This is how we come up with a tolerant compression scheme of small size.

---

### Official Review · Reviewer_MDPr · 2022-07-13

**Rating:** 7
**Confidence:** 4
**Soundness:** 4 excellent
**Presentation:** 3 good
**Contribution:** 2 fair

**Summary:**

This paper introduce adversarial robust learning based on tolerance (i.e., training with larger random perturbation), estimator smoothing and sample compression. It present a rigorous sample complexity analysis. This paper focuses solely on theoretical analysis, and doesn't provide any concrete algorithm or simulation result.

**Questions:**

1. How realistic is the assumption of that "P is V-robustly realizable"? what happens if we let gamma go to infinity?
2. Property 1, the equation should be the expectation of an indicator function.
3. Theorem 10 implicitly requires A_H to be an optimal PAC learning, which is not clearly mentioned.

**Limitations:**

N.A.

**Strengths And Weaknesses:**

1. Strength: the results are mathematically sound.
2. Weakness: despite the rigorous mathematical derivation, I am not sure how the presented PAC analysis result helps the community. One can directly apply the sample complexity analysis on the adversarial loss (say via Rademacher complexity). Does the presented result provide more insights or a better rate?
3. Weakness: the paper doesn't actually provide an executable algorithm, but only an existence results, i.e., the existence of measure mu, the existence of sample compression scheme. Whether the rate results pair with a computational feasible algorithm is unclear, and there is no simulation to demonstrate empirical performance/comparison.
4 Weakness: in conclusion, the paper neither provide a theoretical information limit result, nor an executable adversarial training algorithm.

---

> ### Author Response · Authors · 2022-08-02
> **The Rademacher complexity of the adversarial loss class does not characterize adversarial PAC learning**
>
> >Weakness: despite the rigorous mathematical derivation, I am not sure how the presented PAC analysis result helps the community. One can directly apply the sample complexity analysis on the adversarial loss (say via Rademacher complexity). Does the presented result provide more insights or a better rate?
>
> That is indeed the first question to ask if we want to show PAC bounds for adversarially robust learning. Unfortunately several previous works have shown that the Rademacher complexity of the adversarial loss can be unbounded for simple cases. For example, Montasser et al (http://proceedings.mlr.press/v99/montasser19a/montasser19a.pdf) provide a 1-dimensional example where the Rademacher complexity of the adversarial loss is unbounded and yet adversarially robust learning is trivial to do. Therefore, in general, the Rademacher complexity of the adversarial loss class is not the right measure for characterizing the sample complexity. In contrast, our analysis holds even in general (even in the examples above where the Rademacher complexity of the adversarial loss is unbounded).
>
> >Weakness: the paper doesn't actually provide an executable algorithm, but only an existence results, i.e., the existence of measure mu, the existence of sample compression scheme. Whether the rate results pair with a computational feasible algorithm is unclear, and there is no simulation to demonstrate empirical performance/comparison.
>
> While we are not claiming that our algorithms are empirically competitive as is, note that both of the algorithms are constructive given ERM learners. The only potentially non-constructive step is sampling from a uniform distribution over the perturbation sets, which may be difficult for complicated metric spaces. However, if the underlying space is Euclidean (which it often is in practice) then uniformly sampling from most perturbation sets of interest ($\ell_0, \ell_1, \ell_2$ balls) is straightforward.
>
>
> >How realistic is the assumption of that "P is V-robustly realizable"? what happens if we let gamma go to infinity?
>
> The assumption of robust realizability is only realistic when we have a very rich hypothesis class (think of a large neural network that achieves zero adversarial training loss on a data set) and is not realistic in general.. However, robust realizability is required only for our first algorithm. The compression-based algorithm provided in Section 6 does not assume P to be V-robustly realizable.
>
>
> >Property 1, the equation should be the expectation of an indicator function.
>
> Yes, thanks for pointing out the typo!
>
>
> >Theorem 10 implicitly requires $A_H$ to be an optimal PAC learning, which is not clearly mentioned.
>
> It's not clear what the reviewer means by "optimal" here. Is that the PAC-learner that requires the smallest number of samples? We do not need that for the theorem to hold. The theorem simply requires $A_H$ to be any PAC-learner and the sample complexity bounds of the theorem depend on the sample complexity that $A_H$ is able to achieve.

---

> > ### Comment · Reviewer_MDPr · 2022-08-04
> > **Follow up questions**
> >
> > Thanks for the explanations.
> >
> > About my last question; it seems that theorem 10 is claiming that algorithm $A_H$ has sample complexity $m_H(\epsilon,\delta)=[VC(H)+\log(1/\delta)]/\epsilon$, i.e., $A_H$ is an optimal PAC learner.
> >
> > Also, I do agree with Reviewer Lxpv  that empirical justification is necessary for this paper. Although the authors classify their paper as a theory paper, I consider it is a methodology paper because: (1) the paper propose its own tolerant PAC learning framework. It reduces to the common one when gamma is 0 and the result in the paper fails. (2) within this framework, the sample complexity analysis in this work gives an upper bound, but not the lower bound. And this upper bound is yielded by the algorithms proposed in this paper. In other word, the rate analysis is only meaningful when using the algorithms of this paper. Therefore, empirical justification is necessary to show this algorithm(s) indeed is useful. If not, the rates induced by this algorithm is futile. (It is like "I have a good algorithm, which is shown to be consistent! however unfortunately you cannot use it.")
> >
> > On the other hand, if the paper is claiming, e.g., some minimax lower bound (or optimal) of the sample complexity, then it is a different story, since a lower bound analysis reveals information limit. In such case, I will consider it as a pure theoretical contribution.

---

> > > ### Author Response · Authors · 2022-08-05
> > > **thanks for the follow up**
> > >
> > > We would like to thank the reviewer for engaging with this discussion!
> > >
> > > >About my last question; it seems that theorem 10 is claiming that algorithm $A_H$ has sample complexity…..
> > >
> > > We agree with the reviewer that we should be more precise in the stated bound of Theorem 10. The first equality, stating the sample complexity in terms of $m_H(\epsilon,\delta)$, the sample complexity of $A_H$, is precise as is; for the instantiation of the second equality, there is an implicit optimality assumption (which can be achieved by using the known results in literature); we will clarify this.
> > >
> > >
> > > >Also, I do agree with Reviewer Lxpv that empirical justification is necessary for this paper. Although the authors classify their paper as a theory paper, I consider it is a methodology paper…
> > >
> > >
> > > First of all, we echo what Reviewer FjcX mentioned in their new comment. Studying the adversarial PAC learning problem is an active field in learning theory, with many remaining open questions (such as finding an intuitive combinatorial parameter that characterizes the sample complexity). It is very uncommon to see experiments in this line of work.
> > >
> > > We also want to emphasize various aspects that motivate relaxing the definition of adversarial learning to the tolerant version. Here is our argument:
> > >
> > > **1. Relaxing the problem requirement from the non-tolerant to tolerant adversarial learning is sensible from the user’s point of view**: the choice of the perturbation set is part of the adversarial learning setup and is provided by the user of the algorithm. Now consider a scenario that the user considers a ball of radius 3 to be a good choice. Then, likely, 3.00000000 is not the exact optimal choice, but perhaps a ball of radius 3.0000001 would also be a good choice from the practical perspective. This is discussed in the last paragraph of our intro.
> > >
> > > **2. The perturb-and-smooth method that gives a PAC guarantee in the tolerant setting is closely related to techniques that are actually used in practice**: This is basically what we proved in Theorem 10. Note that as the next point emphasizes, this would have not been possible without relaxing the framework for analysis to the tolerant setting. Therefore, the non-tolerant setting of PAC learning cannot explain the success/usefulness of perturb-and-smooth-type algorithms while the natural tolerant setting can.
> > >
> > > **3.On the other hand, the perturb-and-smooth method does not give a guarantee in the non-tolerant adversarial setup**: consider a simple example that the domain set is the real line, and the perturbation set $u(x)$ is an open interval of radius 1 around $x$, i.e., $u(x)=\{z: |x-z|<1\}$. Let the hypothesis class be the class of all thresholds. Also, let the distribution be supported only on two points: with probability 0.5 it generates $(x,y)=(5,0)$, and with probability $0.5$ it generates $(x,y)=(7,1)$. Note that this distribution is robustly realizable, since we can pick the threshold exactly at $x=6$. However, any other threshold like $x=6.000001$ causes a large adversarial loss. Therefore, if one uses the perturb-and-smooth approach it will find a noisy threshold (due to randomness of perturbations) and will get a large adversarial loss.
> > >
> > > **4. The relaxation to the tolerant setting provides novel insights from the theoretical perspective**: we showed how it allows for the removal of the dual VC dimension in the (tolerant) compression approach; this is something that is not known to be possible in the non-tolerant setting (without stronger, additional assumptions).
> > >
> > > **5. Therefore**, it seems that the tolerant setting is sensible from the user’s perspective, is closer to what is being done in practice, and the analysis in this setting provides novel insights into the structure of the problem leading to simpler bounds. We think this is a strong case for this new setup.

---

### Official Review · Reviewer_Lxpv · 2022-07-19

**Rating:** 4
**Confidence:** 2
**Soundness:** 3 good
**Presentation:** 2 fair
**Contribution:** 3 good

**Summary:**

The paper introduces an algorithm called “tolerant perturb and smooth” and analyzes it with the lens of tolerant PAC learning. The analysis applies to the robustly realizable setting. This analysis is able to tighter sample complexity bounds than previous work.

**Questions:**

Is there an argument that this work has higher significance than I've thought, despite the algorithm itself not working well? For example, is it a step on a path to performing an analysis of one of the methods from the related work section that works well?

**Limitations:**

My main objection to this paper is that I think it does not make the limitations clear enough to an audience that is interested in adversarial examples but not expert in PAC learning theory. I'm concerned that the paper comes across as proving that TPAS "works" as a defense when that's not the case and I think that's not technically what the paper is claiming.

One other limitation I didn't discuss above is Algorithm 1 uses an analytical expectation over x’, rather than a finite number of samples, so the algorithm cannot be implemented in practice for most interesting models. I see that this is discussed at line 135 but I think the discussion should be more prominent, I think the problem is worse than being “potentially expensive” for many models such as neural networks.


**Strengths And Weaknesses:**



	Pros:
		- I don’t have the expertise to check the proofs, but I think the proofs in the paper are non-trivial and are likely to be correct

	Cons:
		- I’m fairly confident that the algorithm TPAS algorithm does not work at all. I implemented the algorithm for linear models on MNIST and it got 5% adversarial test accuracy. That’s worse than random guessing! I didn’t even implement the real adversarial example algorithm that’s aware of the random smoothing, I just generated adversarial examples off a single instantiation of the model on clean examples, so presumably the true adversarial error rate is even worse. Most of what I say in the review is low confidence but I am confident that the algorithm simply does not work.
		- Similar algorithms that do work, as correctly discussed in the related work section, do things like train on more than one noisy sample per example, and use different aggregation metrics, rather than just the simple mean.
		- To be clear, I’m not saying that the PAC analysis in the paper is wrong, or that the paper over claims performance anywhere. It’s possible to have low sample complexity by converging rapidly (as a function of dataset size) to very bad performance. The analysis in the paper is also restricted to the robustly realizable setting, meaning the final performance should be good, but this also restricts the applicability of the work significantly. For example, it means the paper is not actually applicable to my linear MNIST experiment.
		- It’s unclear to me whether there is any interesting data distribution (ImageNet etc) which is known to have a model class that satisfies robust realizability. For MNIST and small max norm balls, robust realizability may be possible, though it’s not totally clear that the norm ball convolved with the true data distribution for one class does not overlap with the norm ball convolved with the true data distribution of another class. Certainly the norm balls centered on the finite MNIST test sets don’t overlap, for the size of norm balls commonly used in the literature. Small convolutional neural networks are able to memorize the (adversarial) test set, so maybe these satisfy robust realizability.
		- I also trained a small CNN using TPAS and it got ~1% accuracy against adversarial examples. Once again, this was with a non-adaptive attacker (not aware of the test time noise smoothing).
			- I’m assuming this result is still compatible with the claims in the paper, but I haven’t spent any time trying to work out how so.
		- Overall, I argue for rejection because I’m concerned that as presented, the paper would be misinterpreted as arguing that TPAS is an effective defense against adversarial examples. The adversarial example topic has repeatedly proved to be prone to false memes spreading rapidly, becoming ingrained, and requiring people working on the topic to waste a lot of time and effort arguing against misconceptions. This could potentially be addressed by substantially revising the paper to be extremely up front about the limitations of TPAS and explaining the interpretation of the PAC theory in a way that makes it clear that the PAC theory does not imply that TPAS would result in good performance or even better than chance performance on, say an ImageNet benchmark. Another way of stating this objection is that, while the paper does not over claim, I feel like it omits too much discussion of (very significant) limitations.
		- I have a second, weaker reason for arguing for rejection, which is simply that NeurIPS is a selective venue, publishing particularly interesting results rather than merely correct results. Due to the poor performance of the TPAS algorithm, I argue that theoretical analysis of the algorithm is not interesting enough to be ranked high enough for NeurIPS acceptance, regardless of the correctness of that analysis.

Comments on the dimensions:
- I don't know enough about PAC theory to assess originality or quality well.
- I argue that the work has poor clarity because it will appear to many readers to endorse TPAS as a defense against adversarial examples, when in fact I think it technically does not claim that this method will achieve good performance.
- I argue that the work has low significance because the algorithm analyzed does not work



One sentence I do think is mistaken: I think this sentence should be revised: “These works provide learning algorithms that guarantee low generalization error in the presence of adversarial perturbations in various settings.” Unless I’m mistaken, these algorithms guarantee that the generalization error is nearly optimal within the hypothesis space, but the optimal generalization error within the hypothesis space for many standard hypothesis spaces (like linear models, or some neural network) could be worse than random guessing.

---

> ### Author Response · Authors · 2022-08-02
> **This is a paper on adversarial PAC learning, not an empirical paper**
>
> As the reviewer states themselves, this is a learning theory paper and does not make claims about any concrete empirical success. Studying adversarially robust learning under the PAC-learning framework is a very active area of research and numerous publications have appeared at highly selective venues studying this topic. For example, Montasser et al's work (that we have cited) proved an initial, worse sample complexity bound for adversarial learning (in the non-tolerant setting) and received a COLT best paper award in 2019. Cullina et al., 2018 and Montasser et al 2020b appeared at NeurIPS. Most of earlier PAC type analysis in the adversarial setting is for algorithms that are highly impractical. Our main contribution is to provide a strong PAC guarantee for an algorithmic framework (perturb and smooth) and is much closer to what is used in practice, and we believe that this may provide a path towards more rigorous analysis for practical methods. The mere fact that the reviewer claims that they have implemented one of our algorithms is interesting, as it is generally infeasible to implement other methods in the literature with PAC-type guarantees.
>
> Many of the concerns raised by the reviewer seem to be ignoring 50% of the paper. Our abstract clearly mentions that we provide two algorithms: 1) a simple algorithm that we call TPaS that works under robust realizability and has an exponential dependence on dimension, 2) a more complicated compression-based algorithm that does not require robust realizability and has a linear dependence on dimension. The reviewer criticizes that the PAC analysis for TPaS requires robust realizability (we agree this is a strong assumption  and have made that clear in the paper) but fails to mention that we also provide a second algorithm that does not require that assumption. First providing analysis for an algorithm that requires realizability and then improving it to the agnostic case is a very common theme in the learning theory literature.
>
> To summarize, *our main contribution is a step towards providing strong (eg PAC type) performance guarantees for commonly used algorithmic techniques (perturb and smooth) in adversarial learning, by introducing the tolerant setting for the analysis*. The task of designing simple algorithms that have a provably small sample complexity has so far been challenging. We believe that adding tolerance to the framework might enable us to push the frontiers. We demonstrate that by providing two algorithms in our paper, 1) an algorithm that we call TPaS that is simple to describe and analyze but requires robust realizability, and 2) a compression-based algorithm (that uses ideas from TPaS) that works in the more general agnostic case and has exponentially better sample complexity than the best known algorithm.
>
>
> Replies to specific comments:
>
> >One sentence I do think is mistaken: I think this sentence should be revised: “These works provide learning algorithms that guarantee low generalization error in the presence of adversarial perturbations in various settings.”
>
> We will make this more precise in the final version by indicating that this is under the assumption that the sample size is sufficiently small based on our upper bounds. However, note that the "generalization error" is often taken to mean the difference in the performance of the model on training data and on test data, and it is that meaning that we intend in the sentence above.
>
> >One other limitation I didn't discuss above is Algorithm 1 uses an analytical expectation over x’, rather than a finite number of samples, so the algorithm cannot be implemented in practice for most interesting models. I see that this is discussed at line 135 but I think the discussion should be more prominent, I think the problem is worse than being “potentially expensive” for many models such as neural networks.
>
> There is also an entire paragraph devoted to discussing this issue on line 302. The number of samples required to get the same answer as if we had calculated the analytical expression accurately does not really depend on the "model", thus it's unclear why the reviewer thinks this will be worse for neural networks. As explained in the paper, the number of samples required to get the same answer as the analytical expression with probability $1-\delta$ is $\Omega(\log 1/\delta)$, which is quite small. The comment about it being "potentially expensive" on line 135 is within the context of the model studied in the previous work discussed in that paragraph, where any extra test-time computation is not allowed.

---

> > ### Author Response · Authors · 2022-08-08
> > **follow up with reviewer Lxpv**
> >
> > Since the author-reviewer discussion period is going to end soon we would like to draw your attention to the response to your post as well as the related discussion between the other reviewers. We will be happy to respond to any follow up questions.

---

> > ### Comment · Reviewer_Lxpv · 2022-08-08
> > **Response to rebuttal**
> >
> > My main concerns about the paper are that:
> > 1) while the claims are *technically* correct, they are presented in a way that is likely to result in misinterpretation. -> the rebuttal seems to double down on this and argue that the current presentation describes the limitations sufficiently ("There is also an entire paragraph...")
> > 2) the algorithm being studied doesn't work at all and performs worse than random guessing against an adversary -> the rebuttal seems to concede this, as far as I can tell
> >
> > "The mere fact that the reviewer claims that they have implemented one of our algorithms is interesting, as it is generally infeasible to implement other methods in the literature with PAC-type guarantees." -> obviously I'm not doing the analytical expectation, etc. I thought that making an experiment of an approximate version and giving it the chance to succeed would be preferable to just saying that the algorithm doesn't work a priori.

---

> > > ### Author Response · Authors · 2022-08-09
> > > **Open to clarify if specific pointers to alleged misleading statements would be provided**
> > >
> > > >while the claims are technically correct, they are presented in a way that is likely to result in misinterpretation.
> > >
> > > It is not easy to respond to a blanket statement like this. We would appreciate it if you could elaborate on the *specific examples* of sentences or claims that can be (or are likely to be in your view) misinterpreted and we will be happy to make edits to address them. We had pointed out the parts in the paper where specific concerns from your previous comment have been addressed.
> > >
> > > >the rebuttal seems to double down on this and argue that the current presentation describes the limitations sufficiently ("There is also an entire paragraph...")
> > >
> > > In your previous comment, you had asked about the smoothing step and had mentioned that “I see that this is discussed at line 135”. That is why we thought you have missed the paragraph starting in line 302 that we have dedicated to this topic. We will be happy to elaborate more about this in the intro and link to this paragraph if that’s what you mean.

---

### Official Review · Reviewer_xynd · 2022-09-10

**Rating:** 4
**Confidence:** 3
**Soundness:** 2 fair
**Presentation:** 3 good
**Contribution:** 2 fair

**Summary:**

The paper studies the PAC learnability of adversarial robustness in the tolerant setting. The paper reveals a learnability result showing that VC hypothesis sets are tolerantly PAC learnable in the realizable case using a ``perturb-and-smooth'’ algorithm. As compared to previous non-tolerant results, their sample complexity has an exponential improvement on the dependence of VC dimension. The paper also reveals a learnability result in the agnostic setting through sample compression.


**Questions:**

See the weaknesses.

**Limitations:**

The discussion was adequate.

**Strengths And Weaknesses:**

Strengths:

* Well-written assumptions and theorems.
* New tolerant framework is considered.

Weaknesses:

The significance is limited. The major issue is that their tolerance framework is not justified.

On the one hand, they do not specify how to choose a reasonable *reference* perturbation type V(x) given an *actual* perturbation type U(x) in definition 4. More precisely, the choice of tolerance parameter \gamma in line 206 is unclear. I think this is very crucial and necessary in this framework. Consider the case where V(x)=0 and U(x) = B_{\gamma}(x). Then the inequality between line 195 and 196 tells that the algorithm can find a classifier with clean error smaller than the adversarial bayes error within ball B_{\gamma}(x), which is actually meaningless for any \gamma since we know that the clean bayes error must be less than the adversarial bayes error and the sample-efficient PAC learners for clean bayes classifier is known.

In line 298 - 301, they want to set gamma to be above some threshold. Using their parameters of \zeta=1, c =1000 and d=1000 in the provided example, the gamma is required to be larger than 1. It is hard to believe that such a large gamma is meaningful in terms of the tolerance of perbutation size.

On the other hand, they do not show negative results like "some tolerance is necessary otherwise adversarial robustness is not possible", which further limits the significance.

Their sample complexity such as that in Theorem 10 is independent of the actual perturbation size r for adversarial robustness. Their proposed algorithm does not train on the worst case perturbed samples. Instead, the algorithm only trains on the randomly perturbed samples, which I think is not useful for adversarial robustness. These facts suggest that such a tolerance framework seems not reliable.

---

### Comment · Reviewer_FjcX · 2022-08-04
**Disagreeing with the other reviewers**

Hi,

after reading the other two reviews, I would like to describe my point of view and insist that this paper should be accepted.
I'm very familiar with the theoretical results in this field.
This paper makes a clean and nice improvement in the PAC model with adversarial examples.

I don't think that experiments are necessary for this paper.
Also, not every paper provides sharp upper and lower bounds. What should be considered is the contribution of the paper to the field, new ideas, and analysis.
Many questions in this field are open, we don't even know the right parameter characterizing this model
(also under robust realizability assumption!), therefore we cannot hope for sharp results at this point.

I don't find good arguments in the other reviews for rejecting the paper.
I believe that this paper should be accepted, and at the least get the AC's attention.

---

> ### Comment · Reviewer_MDPr · 2022-08-04
> **a question to reviewer FjcX**
>
> Hi, FjcX,
>
> Thanks for the feedback. I must confess that I am not familiar with the recent development of PAC learning in adversarial training and I feel that I am too harsh on the rating given that this is the first PAC result in the fielding. I am considering raising my score, but would like to ask the expert reviewer FjcX one more question: is it common in the PAC learning community that no simulation is conducted?

---

> > ### Comment · Reviewer_FjcX · 2022-08-06
> > **Discussion with MDPr**
> >
> > Hi,
> >
> > yes, it is very common not including experiments for these types of results of sample complexity in a PAC setting.
> > I will explain why I don't think that experiments would be useful here. The results in this paper hold for a general case of any function class and any perturbation set that is a ball in some metric space (capturing the interesting case of Euclidean space).
> > If the results were more limited, maybe conducting experiments to show a more general phenomenon would be enlightening in my point of view.
> >
> > I would like to give the context of this paper to the current literature.
> > For the setting of any function class and any perturbation set (not necessarily in a metric space),
> > we know that a finite VC dimension is sufficient for learning, with a sample complexity of roughly VC(H)2^VC(H).
> > We also know that it is not necessary, there exists a class with infinite VC that is learnable, as opposed to the standard PAC setting with 0-1 loss. We are aware of some other dimensions that play a role (in the lower bound), but this is an open question to characterize learnability in this setting.
> >
> > I think that this paper contributes to this line of work.
> > Assuming a perturbation set in a metric space is quite general and captures interesting scenarios.
> > We should note that the tolerance parameter (gamma) is only in a log factor in Thm 2.
> > Also, given a known reduction from agnostic the realizable, the results in this paper hold for the agnostic case as well.

---

> ### Comment · Reviewer_Lxpv · 2022-08-08
> **Open to discussion**
>
> I'm open to discussing my rating. I don't think your comment address my reasons for a low score.
>
> "I don't think that experiments are necessary for this paper." : Neither do I. I ran some experiments of my own as part of my review to back up my claim that the proposed algorithms are worse than random guessing. I don't think the paper needs to have experiments added to it.
>
> My 2 main reasons for rejecting are:
>
> 1) I'm concerned that people interested in adversarial examples but not expert in PAC learning would take this paper as an endorsement of TPaS as an effective defense against adversarial examples, when it is in fact worse against an adversary than random guessing. Even the title "Adversarially Robust Learning..." implies that the algorithm does become robust in some way. For this reason, I'm arguing for rejection in terms of an absolute standard, I don't want to endorse something that I think could cause widespread misconceptions. I think in this respect you're looking at this coming from the PAC community, while I'm looking at this coming from the adversarial example community, which has a significant problem with incorrect beliefs about defense effectiveness spreading widely.
>
> 2) I think that because the algorithm does not work at all, the analysis is not very interesting. In this case I'm arguing for rejection on a relative standard---the AC probably has something like 12 papers and is meant to pick something like 3 to accept, so I'm signaling that a technically correct paper on an uninteresting topic probably doesn't rank in the top ~3 of the ACs stack. I realize that other PAC analysis in the adversarial setting have been accepted before; I wasn't a reviewer of those papers. Speaking as someone who is not part of the PAC learning theory, I believe that PAC analysis has been interesting in the past when it analyzed algorithms that do actually work, such as supervised learning in the non-adversarial setting. I don't understand why anyone should care about the sample complexity of something that does not work---low sample complexity just means it converges to bad performance fast. While this is my secondary reason for rejection, it is one where I feel like FjcX might be able to persuade me (by explaining why it's interesting).

---

> > ### Author Response · Authors · 2022-08-09
> > **Aiming to provide some more clarifcation, willing to respond to concrete suggestions, but not to blanket criticism**
> >
> > >Even the title "Adversarially Robust Learning..." implies that the algorithm does become robust in some way.
> >
> > This paper introduces a new formulation for defining the success of an adversarial learning method through the novel definition of learning with tolerance. The title does not suggest any specific method or algorithm; it is a reference to the theoretical problem statement. ”Adversarially robust learning” has been used as a name for this problem statement in previous work. The two methods that we propose address that problem statement.
> >
> >
> > >I don't want to endorse something that I think could cause widespread misconceptions. I think in this respect you're looking at this coming from the PAC community, while I'm looking at this coming from the adversarial example community, which has a significant problem with incorrect beliefs about defense effectiveness spreading widely.
> >
> > We have supported our claims with clear definitions, accurate theorem statements, and rigorous proofs. This is the style of many papers published in NeurIPS. There is no need for the reader to be an expert in PAC learning to understand the meaning of theorems or claims. Again, we will be happy to address if there are specific parts of the paper that you found misleading.
> >
> >
> > >the algorithm being studied doesn't work at all and performs worse than random guessing against an adversary
> >
> > Arguing that an algorithm “doesn’t work at all” based on the reviewer’s own implementation and choice of parameters/task/data-set is quite odd. The reviewer also seems to ignore our previous response that we have presented two methods: TPaS (which requires robust realizability and has an exponential sample complexity in terms of the doubling dimension) and the compression-based approach that addressed both of these issues.
> >
> >
> > >Speaking as someone who is not part of the PAC learning theory, I believe that PAC analysis has been interesting in the past when it analyzed algorithms that do actually work, such as supervised learning in the non-adversarial setting. I don't understand why anyone should care about the sample complexity of something that does not work
> >
> > Such a blanket statement undermines a large amount of the developments in the learning theory community. PAC learning helps to understand the limits of learning in a scenario (i.e., upper bounds and lower bounds). It also offers practical insights. For example, our compression-based result states that it is possible to learn a hypothesis class with near-linear dependence on its VC dimension. This was not known to hold, and offers new insights on the limits of learnability. Our TPaS method signifies the importance of studying the tolerant setting (see our response to Reviewer MDPr’s follow up).

---

> > > ### Comment · Reviewer_FjcX · 2022-08-09
> > > **I completely agree**
> > >
> > > I completely agree and couldn't phrase it better.

---

### Meta-Review · Area_Chair_WrXw · 2022-08-23

**Recommendation:** Reject
**Confidence:** Less certain

**Metareview:**

The paper introduces an algorithm called “tolerant perturb and smooth” and analyzes it with the lens of tolerant PAC learning. The analysis applies to the robustly realizable setting and achieves tighter sample complexity bounds than previous work.

However, there were concerns about the significance. The major issue is that their tolerance framework is not justified.
From the discussion:
<<
On the one hand, they do not specify how to choose a reasonable reference perturbation type U(x) given an actual perturbation type U(x) in definition 4. More precisely, the choice of tolerance parameter \gamma in line 206 is unclear. I think this is very crucial and necessary in this framework. Consider the case where V(x)=0 and U(x) = B_{\gamma}(x). Then the inequality between line 195 and 196 tells that the algorithm can find a classifier that has clean error smaller than that of adversarial bayes error within ball B_{\gamma}(x), which is actually meaningless for any \gamma since we know that the clean Bayes error must be less than that of adversarial Bayes error and the sample-efficient PAC learners for clean Bayes classifier is known.

In line 298 - 301, they want to set gamma to be above some threshold. Using their parameters of \zeta=1, c =1000 and d=1000 in the provided example, the gamma is required to be larger than 1. It is hard to believe that such a large gamma is meaningful in terms of the tolerance of perturbation size.

On the other hand, they do not show negative results like "some tolerance is necessary otherwise adversarial robustness is not possible", which further limits the significance.

Their sample complexity such as that in Theorem 10 is independent of the actual permutation size r for adversarial robustness. Their proposed algorithm does not train on the worst-case perturbed samples. Instead, the algorithm only requires training on the randomly perturbed samples, which I feel is not useful for adversarial robustness. It will be surprising if such a tolerance framework is reliable based on these facts.
>>

The referees raised serious concerns regarding the efficacy and significance of the algorithm. Given the lengths to which they went in running independent experience, I take these reservations very seriously and cannot recommend acceptance at this stage.

**Award:**

No

---

### Decision · Program_Chairs · 2022-09-14

Reject